# Crawling toward obsolescence: The extended lifespan of amylase for pancreatitis

**Naga Sasidhar Kanaparthy**[1,2], **Andrew J. Loza**[1,2], **Ronald George Hauser**[2,3]*

**1** Department of Emergency Medicine, Yale University School of Medicine, New Haven, Connecticut, United States of America, **2** Veterans Affairs Connecticut Healthcare, West Haven, Connecticut, United States of America, **3** Department of Laboratory Medicine, Yale University School of Medicine, New Haven, Connecticut, United States of America

☯ These authors contributed equally to this work.

* ronald.hauser@yale.edu

**Data Availability Statement:** All relevant data are within the paper and its Supporting information files.

**Funding:** The author(s) received no specific funding for this work.

## Abstract

The correlation between hyperamylasemia and acute pancreatitis was discovered in 1929, yet another test, lipase, was shown to provide better diagnostic performance in the late 1980s and early 1990s. Subsequent studies demonstrated co-ordering amylase with lipase did not provide additional benefit, only added cost. We sought to investigate the impact of studies advocating for the obsolescence of amylase on its clinical demand. We reviewed 1.3 million reportable results for amylase over 14 years (2009–2022). The trend in utilization of amylase over this period declined by 66% along a linear trajectory (R2 = 0.97). Despite demand for amylase decreasing by an average of 17,003 tests per year, the last year of the study (2022) recorded over 100,000 results for amylase. By interpolating the decline of amylase until the utilization reached zero, we calculated amylase orders will continue for 6 more years until 2028. Tests for creatinine and lipase changed <3% over the same period. Despite a multitude of studies advocating for the obsolescence of amylase, robust demand continues. Many important clinical guidelines, a source many practicing physicians rely on, have yet to acknowledge the preference for lipase over amylase. They frequently treat the two tests as equivalent, neglecting their head-to-head comparison studies and subsequent studies advocating against co-ordering both tests simultaneously. To expedite the obsolescence of amylase, which we anticipate lasting 46 years in our case study from its initial call for obsolescence to the last orders placed, metrics created specifically to monitor the utilization of unnecessary tests are also needed.

## Background

Fueled by tens of billions of dollars in research funds a year, healthcare evolves through the introduction of innovative technology: new medications, therapeutic procedures, and diagnostics [1]. These advances permeate clinical guidelines. With time clinical practice evolves, and old methods become obsolete. In the evolutionary history of diagnostic tests CK-MB, as an example, has become outdated with the introduction of troponins T and I [2]. Many tests have met a similar fate, as documented by the Centers for Medicare & Medicaid Services' list of

**Competing interests:** The authors have declared that no competing interests exist.

"Obsolete or Unreliable Diagnostic Tests" [3]. Instead of focusing on the evolution of the medicine through innovation, we take an opposite perspective, documenting the natural history of a laboratory test, pancreatic amylase, as changes in scientific evidence slowly relegate this test to the past.

In 1811 the discovery of a mysterious substance that could split starch occurred [4]. It initially received the named "diastase" from the Greek word "diastasis" meaning separation. This discovery would begin the field of enzymology, with the suffix "-ase" becoming standard nomenclature for the naming of enzymes. Later amylase, from the Greek work "amylon", which means starch, was identified as the catalyst of this reaction [5]. Over 100 years later in 1929, the correlation between hyperamylasemia and acute pancreatitis became known to clinical medicine [6]. By 1972, an article in the New England Journal of Medicine (NEJM) wrote, "The serum amylase remains the most important single determination of acute pancreatitis" [7]. Thus, at this point amylase became the standard test for diagnosis of pancreatitis.

But despite reaching its zenith of clinical utility, limitations in the diagnostic ability of amylase for the diagnosis of acute pancreatitis were apparent. Multiple organs produce amylase; it is not specific to the pancreas. In an attempt to overcome this limitation, further research identified two principal isoenzymes: p-type and s-type isoamylase. P-type isoamylase originates from the pancreas, and s-type isoamylase originate from a variety of tissues including the salivary glands [4]. Although this discovery had the potential to improve the specificity of amylase, modern assays still rely on the catalytic ability of amylase, a property both isoenzymes possess [8, 9]. Thus, as pointed out in 1976, the absence of a "simplified routine analytic procedure" that can differentiate the isoenzymes limits the specificity of amylase for acute pancreatitis [5].

Clinical studies would add support to another diagnostic test, serum lipase. Lott et al in 1991 suggested that lipase should replace amylase [10]. In multiple head-to-head comparisons between serum amylase and lipase for the diagnosis of pancreatitis, lipase had superior test characteristics [10–13]. Due to the longer half-life and higher peak concentrations of lipase as compared to amylase, lipase proved to be a superior test, especially in cases where the diagnosis was delayed [13, 14]. Lipase performed better in acute alcoholic pancreatitis, a subtype of acute pancreatitis [12]. False elevations from macrocomplexes appeared less often with lipase than amylase [15]. In stark contrast to the declaration in the NEJM of the superiority of amylase in 1972, which did list lipase as a contender with amylase, the conclusion of a 1991 study in Clinical Chemistry directly comparing amylase to lipase flatly stated, "Amylase is a poor test in the diagnosis of pancreatitis; [a] better choice would be the lipase test. . ." [10].

One would presume the lifespan of amylase drew closer to obsolescence as the preference for lipase over amylase permeated clinical guidelines for pancreatitis. Beginning in 2005 with a pragmatic United Kingdom working group, lipase became the test of choice for pancreatitis instead of amylase [16]. A reasonable hypothesis would postulate that other guidelines would soon follow suit, and amylase would quickly disappear thereafter. To investigate this question, we reviewed the large-scale utilization of amylase over many years.

## Methods

We reviewed reportable results from the Veterans Health Administration (VHA). The VHA is the largest integrated healthcare system in the United States serving between 6 and 7 million Veterans annually. It operates 130 healthcare system in all 50 US states. The data for the study spans from 2009 to 2022, a total of 14 years. As the nation's largest provider of graduate medical education, the utilization of amylase within the VHA likely approximates the utilization of amylase throughout the United States.

To control for changes in laboratory test utilization during the study period, we also reviewed the reportable results for creatinine and lipase. To our knowledge, VA policy during the study period did not provide guidance for or against the utilization of amylase in the workup of acute pancreatitis or any other medical condition.

## Results

From its peak in 2009, amylase testing decreased by 66% (311,752 to 106,503) in 2022 (14 years). The decline followed a linear pattern from 2010 onwards ($R^2 = 0.97$) (Fig 1). The most notable variation occurred around the COVID-19 pandemic (2019–2021). Compared to amylase, laboratory utilization of creatinine and lipase remained steady. From the first to the last year of the 14 year study period, creatinine testing changed by only 2.78% (12,096,416 to 11,759,568) and lipase changed by 1.86% (303,570 to 297,920) (S1 Table). At the current trajectory, amylase will become obsolete in 2028 (6 years), making its total period of decline in reportable results equal to 20 years (2009–2028 inclusive).

## Discussion

We present evidence in the form of over 3.1 million amylase results to demonstrate the prolong time to obsolescence for amylase, estimated at 43 years (1986 to 2028) in our case example. The beginning of amylase's obsolescence occurred in 1986 when Lott et. al. wrote, "Serum lipase determinations with the current, simpler technology are superior to total amylase in the diagnosis of patients with acute pancreatitis" [11]. Amylase appears poised to continue to live into the future, with an additional 6 more years or until 2028. Despite its loss in the head-to-head comparison studies with lipase, amylase has enjoyed an extended lifespan.

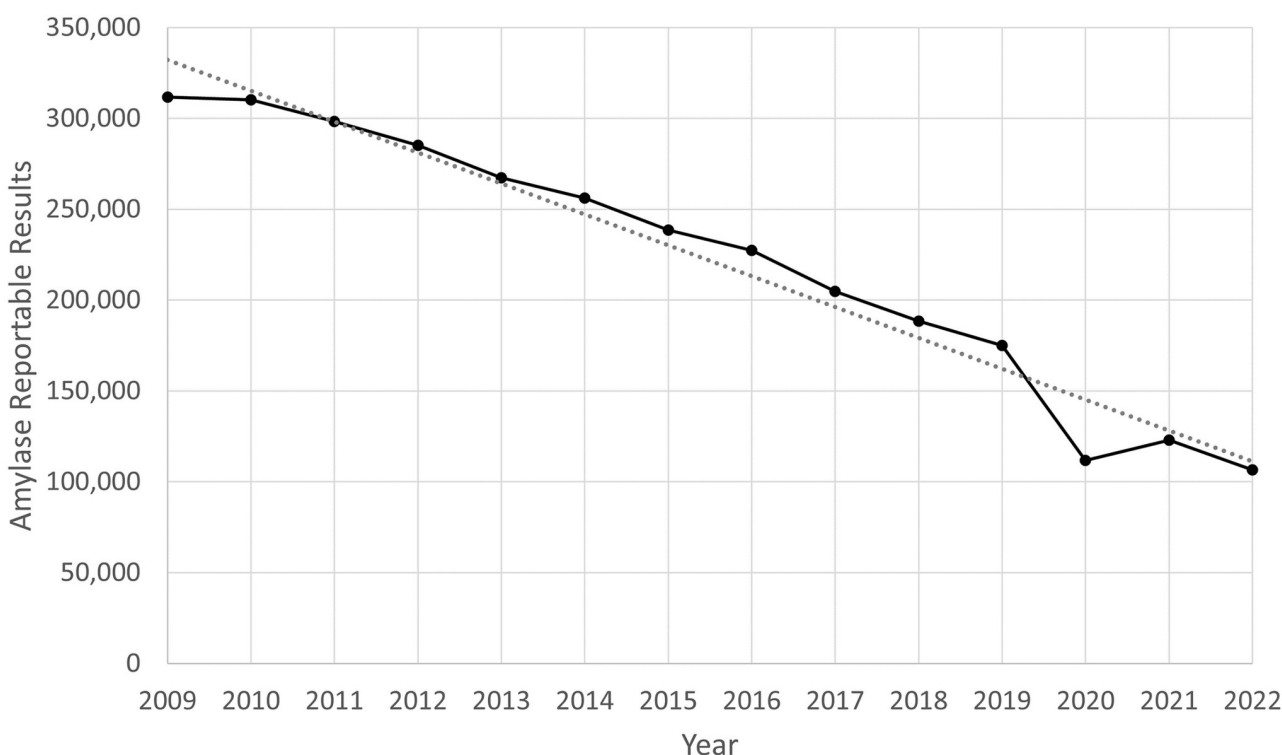

**Fig 1. Amylase reportable results (2009–2022).** The gray dotted line is a linear fit ($R^2 = 0.97$, slope = -17,003).

Co-ordering of lipase and amylase is not recommended. In a literature review of this question, the authors concluded co-ordering had "shown little to no increase in the diagnostic sensitivity and specificity" [17]. The ordering of both tests contributed to unnecessary laboratory expenditures, a position further supported by the Choosing Wisely campaign to reduce unnecessarily diagnostic test [18]. Multiple contemporary utilization reviews lend further support to this idea [19–21].

Guidelines and reviews of acute pancreatitis frequently neglect the comparison studies of lipase and amylase, indirectly promoting the continued utilization of amylase. For example, the American Gastroenterological Association Institute Guidelines state the diagnosis of acute pancreatitis requires, "biochemical evidence of pancreatitis (ie, amylase or lipase. . .)". Amylase and lipase are listed in alphabetical order without clarification of their performance. However, the citation from which this statement is derived, the Atlanta classification of acute pancreatitis, states "serum lipase activity (or amylase activity)", implying a preference for lipase by referring to amylase in the parenthetical. But neither go into any detail about the choice of diagnostic, despite its importance. This oversight in differentiating lipase from amylase in clinical guidelines and reviews of acute pancreatitis has likely led to the continued utilization of amylase despite its inferiority to lipase (Table 1).

Our study has limitations. We present data from a large, geographically diverse healthcare system responsible for teaching many future clinicians, but these findings may not generalize to other healthcare systems, especially the estimated time to obsolescence. Rather than argue for a specific obsolescence timeframe, we hope to make the point that amylase has outlasted its prime by more years than a reasonable person would assume. Amylase utilization may never reach zero, as alternative indications for amylase exist, including salivary and ovarian tumors, ulcer, celiac disease, pregnancy, burns, and certain medications. Pancreatitis represent the most

**Table 1. Recognition of lipase as a superior test to amylase in guidelines and reviews of acute pancreatitis.** Many studies appear to misquote the 2012 Revised Atlanta Classification placing amylase before lipase, thus neglecting the preference for lipase.

| Guideline / Review | Text Quote(s) | Preference for Lipase? | Alphabetical order: Amylase then Lipase |
|---|---|---|---|
| Atlanta Classification (1992) [22] | "elevated pancreatic enzyme levels in the blood and/or urine" | No | Not mentioned |
| AGA Institute Medical Position Statement on Acute Pancreatitis (2007) [23] | "The diagnosis should be based on compatible clinical features and elevations in amylase or lipase levels. . . Elevation of lipase levels is somewhat more specific and is thus preferred." | Yes | Yes |
| Revised Atlanta Classification (2012) [24] | "two of the following three features: . . . (2) serum lipase activity (or amylase activity) at least three times greater than the upper limit of normal. . ." | Yes | No |
| IAP/APA Guidelines (2013) [25] | "The definition of acute pancreatitis is based on the fulfillment of '2 out of 3' of the following criteria: . . . laboratory (serum amylase or lipase >3x upper limit of normal) . . ." | No | Yes |
| AGA Institute Guideline on Initial Management of Acute Pancreatitis (2018) [26] | "The diagnosis of [acute pancreatitis] requires at least 2 of the following features: . . . biochemical evidence of pancreatitis (ie, amylase or lipase elevated >3 times the upper limit of normal). . ." | No | Yes |
| Current Opinion in Gastroenterology (2018) | "serum amylase, and/or lipase greater than three times the upper limit of normal" | No | Yes |
| Nature Reviews Gastroenterology & Hepatology(2019) [27] | "serum amylase and/or lipase elevation more than three times the upper limit of normal" | No | Yes |
| Annals of Internal Medicine (2021) [28] | "Elevation of serum amylase and/or lipase levels to at least 3 times the upper limit of normal is a key component of diagnosing acute pancreatitis." | No | Yes |
| JAMA (2021) [29] † | "Standard chemistries with amylase, lipase, and liver panel tests can help confirm the diagnosis of acute pancreatitis" | No | Yes |

Abbreviations: American Gastroenterologic Society (AGA), International Association of Pancreatology (IAP), American Pancreatic Association (APA), Journal of the American Medical Association (JAMA).

† Authors of this review were criticized for not noting the superiority of lipase and advocating for a lipase-only policy [30].

common reason for amylase testing, a practice identified as "a thing we do for no reason" [21]. We should, as providers of medicine, continue to think about innovative ways to fast track the adoption of evidence such as the creation of metrics specific to the utilization of tests and award programs for adherence to these metrics administered by groups such as Choosing Wisely.

## Conclusion

Diagnostic tests play an important supporting role in medicine. In the case of amylase, the detailed studies conducted in the late 1980s and early 1990s comparing lipase to amylase have not yet received their proper integration into clinical guidelines for acute pancreatitis. This has likely prolonged the lifespan of amylase in the workup of acute pancreatitis and led to additional, unnecessary costs.

## Supporting information

**S1 Table. Yearly totals.** Yearly totals of amylase, creatinine, and lipase from 2009 to 2022. (XLSX)

## Author Contributions

**Conceptualization:** Naga Sasidhar Kanaparthy, Ronald George Hauser.

**Formal analysis:** Andrew J. Loza, Ronald George Hauser.

**Methodology:** Naga Sasidhar Kanaparthy, Andrew J. Loza, Ronald George Hauser.

**Resources:** Naga Sasidhar Kanaparthy, Ronald George Hauser.

**Software:** Ronald George Hauser.

**Supervision:** Naga Sasidhar Kanaparthy, Ronald George Hauser.

**Validation:** Naga Sasidhar Kanaparthy, Andrew J. Loza, Ronald George Hauser.

**Visualization:** Ronald George Hauser.

**Writing – original draft:** Naga Sasidhar Kanaparthy, Andrew J. Loza, Ronald George Hauser.

**Writing – review & editing:** Naga Sasidhar Kanaparthy, Andrew J. Loza, Ronald George Hauser.

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
