## [Decision Letter · Decision Letter 0]

20 Nov 2023

PONE-D-23-33858Crawling Toward Obsolescence: The Extended Lifespan of AmylasePLOS ONE

Dear Dr. Hauser,

Thank you for submitting your manuscript to PLOS ONE. After careful consideration, we feel that it has merit but does not fully meet PLOS ONE’s publication criteria as it currently stands. Therefore, we invite you to submit a revised version of the manuscript that addresses the points raised during the review process.

We look forward to receiving your revised manuscript.

Kind regards,

Amit Ranjan, Ph.D.

Academic Editor

PLOS ONE

Journal Requirements:

Reviewers' comments:

Reviewer's Responses to Questions

**Comments to the Author**

1. Is the manuscript technically sound, and do the data support the conclusions?

Reviewer #1: Partly

Reviewer #2: Partly

2. Has the statistical analysis been performed appropriately and rigorously? 

Reviewer #1: Yes

Reviewer #2: N/A

3. Have the authors made all data underlying the findings in their manuscript fully available?

Reviewer #1: Yes

Reviewer #2: No

4. Is the manuscript presented in an intelligible fashion and written in standard English?

Reviewer #1: Yes

Reviewer #2: Yes

5. Review Comments to the Author

Reviewer #1: I do agree that evaluative studies at times, do not require IRB if there is no interaction or intervention with human subjects and or did not include any identifiable private information.

It is also, well noted that ordering for amylase as an adjunct to lipase in acute pancreatitis creates extra cost. However, increased amylase is seen in quite a number of conditions e.g., intestinal disease, pancreatic disease, decreased metabolic clearance and salivary disease. Thus, crawling toward obsolescence can perhaps be tagged to pancreatitis alone and not other notable conditions. At least, authors should insert a statement or two for this clarification.

I do agree with statements on lines 63 to 70. But again, reading through, it seems amylase obsolescence is just tagged to acute pancreatitis. Thus, prior statements are needed to consolidate this obsolescence in only acute pancreatitis.

Reviewer #2: The article is insightful and a good read. the following points are recommended:

1. The authors are advised to check if the abstract format and referencing style is in line with the format of PLOS ONE.

2. Line 50 check grammar - 'changes in' instead of 'changes to'.

3. Line 59 information on the serum amylase test could be added.

4. Line 72 more information on the serum lipase test could be added.

5. Line 74-75 can be reframed for clarity.

6. Line 76 the term 'subpopulation' may be replaced by 'subset' or other suitable term.

7. Line 80 kindly check the relevance of [a] in the sentence.

8. Line 107 The authors are requested to input the data for the utilization of creatinine and lipase over the period of study

6. PLOS authors have the option to publish the peer review history of their article (what does this mean?). If published, this will include your full peer review and any attached files.

---

## [Author Response · Author response to Decision Letter 0]

6 Dec 2023

Journal Requirements: 

- We have updated the style requirements as per the template in the title page. 

- We have updated the style requirements in the body of the manuscript. Font sizes of headings have been changed 18.

- The figure image has been removed from manuscript and will be uploaded separately. We kept the title and legend at the end of the Results section.

2. In your Data Availability statement, you have not specified where the minimal data set underlying the results described in your manuscript can be found. PLOS defines a study's minimal data set as the underlying data used to reach the conclusions drawn in the manuscript and any additional data required to replicate the reported study findings in their entirety. All PLOS journals require that the minimal data set be made fully available

- We will submit an Excel document as a supplemental document. This file contains the raw aggregate data used for this study.

We have reviewed the reference list for accuracy. There are no retracted articles that have been cited here. The references have been listed in chronological order of appearance in the body of the article. The Endnote library has been been cross-checked. The references are in Vancouver style.

 

Reviewers' comments:

Reviewer's Responses to Questions 

Comments to the Author

1. Is the manuscript technically sound, and do the data support the conclusions?

Reviewer #1: Partly

Reviewer #2: Partly

2. Has the statistical analysis been performed appropriately and rigorously? 

Reviewer #1: Yes

Reviewer #2: N/A

3. Have the authors made all data underlying the findings in their manuscript fully available?

Reviewer #1: Yes

Reviewer #2: No

- We will submit an Excel document as a supplemental document. This file contains the raw data used for this study. 

4. Is the manuscript presented in an intelligible fashion and written in standard English?

Reviewer #1: Yes

Reviewer #2: Yes

5. Review Comments to the Author

Reviewer #1: I do agree that evaluative studies at times, do not require IRB if there is no interaction or intervention with human subjects and or did not include any identifiable private information.

It is also, well noted that ordering for amylase as an adjunct to lipase in acute pancreatitis creates extra cost. However, increased amylase is seen in quite a number of conditions e.g., intestinal disease, pancreatic disease, decreased metabolic clearance and salivary disease. Thus, crawling toward obsolescence can perhaps be tagged to pancreatitis alone and not other notable conditions. At least, authors should insert a statement or two for this clarification.

I do agree with statements on lines 63 to 70. But again, reading through, it seems amylase obsolescence is just tagged to acute pancreatitis. Thus, prior statements are needed to consolidate this obsolescence in only acute pancreatitis.

- We agree with the reviewer, and, in response, we have modified the title to indicate amylase utilization in the context of pancreatitis. 

o “Crawling toward obsolescence: The extended lifespan of amylase for pancreatitis.”

- We agree with the importance for distinguishing amylase ordering for pancreatitis vs. other conditions. As such we have modified the discussion to indicate this.

o Amylase utilization may never reach zero, as alternative indications for amylase exist, including salivary and ovarian tumors, ulcer, celiac disease, pregnancy, burns, and certain medications. Pancreatitis represent the most common reason for amylase testing, a practice identified as “a thing we do for no reason”

Reviewer #2: The article is insightful and a good read. the following points are recommended:

1. The authors are advised to check if the abstract format and referencing style is in line with the format of PLOS ONE.

-The abstract format has been adjusted 

2. Line 50 check grammar - 'changes in' instead of 'changes to'. - 

Thank you for pointing this out. Change has been made 

3. Line 59 information on the serum amylase test could be added.

- added a line at 61 for clarity: Thus, at this point amylase became the standard test for diagnosis of pancreatitis.

4. Line 72 more information on the serum lipase test could be added.

- added a line at 74 -75 to signify the change of guard from amylase to lipase – “Lott et al in 1991 suggested that lipase should replace amylase”

5. Line 74-75 can be reframed for clarity. 

- We have rephrased this sentence for better clarity (Line 75-78)

Due to the longer half-life and higher peak concentrations of lipase as compared to amylase, lipase proved to be a superior test, especially in cases where the diagnosis was delayed.

6. Line 76 the term 'subpopulation' may be replaced by 'subset' or other suitable term.

 - replaced subpopulation with “subtype” (line 79)

7. Line 80 kindly check the relevance of [a] in the sentence.

Response: We added the square brackets to signify a clarification to the quote. See the APA style manual. 

8. Line 107 The authors are requested to input the data for the utilization of creatinine and lipase over the period of study

Response: From the first to the last year of the 14 year study period, creatinine testing changed by only 2.78 % (12,096,416 to 11,759,568) and lipase changed by 1.86% (303,570 to 297,920).

6. PLOS authors have the option to publish the peer review history of their article (what does this mean?). If published, this will include your full peer review and any attached files.

Fig 1 was uploaded and checked on the PACE too. The new file with a reduced size has been uploaded. Please note the initial file was .tiff and it was changed to .tif

---

## [Editor Report · Decision Letter 1]

8 Dec 2023

Crawling toward obsolescence: The extended lifespan of amylase for pancreatitis

PONE-D-23-33858R1

Dear Dr. Hauser,

We’re pleased to inform you that your manuscript has been judged scientifically suitable for publication and will be formally accepted for publication once it meets all outstanding technical requirements.

Kind regards,

Amit Ranjan, Ph.D.

Academic Editor

PLOS ONE
---

## [Editor Report · Acceptance letter]

12 Dec 2023

PONE-D-23-33858R1 

Crawling toward obsolescence: The extended lifespan of amylase for pancreatitis 

Dear Dr. Hauser:

I'm pleased to inform you that your manuscript has been deemed suitable for publication in PLOS ONE. Congratulations! Your manuscript is now with our production department. 

Kind regards, 

on behalf of

Dr. Amit Ranjan 

Academic Editor

PLOS ONE